# Bone marrow central memory and memory stem T-cell exhaustion in AML patients relapsing after HSCT

Maddalena Noviello[1], Francesco Manfredi[1], Eliana Ruggiero[1], Tommaso Perini[2], Giacomo Oliveira[1], Filippo Cortesi[3], Pantaleo De Simone[1], Cristina Toffalori[4], Valentina Gambacorta[4], Raffaella Greco[2], Jacopo Peccatori[2], Monica Casucci[5], Giulia Casorati[3], Paolo Dellabona[3], Masahiro Onozawa[6], Takanori Teshima [6], Marieke Griffioen[7], Constantijn J.M. Halkes[7], J.H.F. Falkenburg[7], Friedrich Stölzel[8], Heidi Altmann[8], Martin Bornhäuser[8], Miguel Waterhouse[9], Robert Zeiser[9], Jürgen Finke[9], Nicoletta Cieri[1,11], Attilio Bondanza[5], Luca Vago[4], Fabio Ciceri[2,10] & Chiara Bonini[1,10]

The major cause of death after allogeneic Hematopoietic Stem Cell Transplantation (HSCT) for acute myeloid leukemia (AML) is disease relapse. We investigated the expression of Inhibitory Receptors (IR; PD-1/CTLA-4/TIM-3/LAG-3/2B4/KLRG1/GITR) on T cells infiltrating the bone marrow (BM) of 32 AML patients relapsing (median 251 days) or maintaining complete remission (CR; median 1 year) after HSCT. A higher proportion of early-differentiated Memory Stem ($T_{SCM}$) and Central Memory BM-T cells express multiple IR in relapsing patients than in CR patients. Exhausted BM-T cells at relapse display a restricted TCR repertoire, impaired effector functions and leukemia-reactive specificities. In 57 patients, early detection of severely exhausted (PD-1$^+$Eomes$^+$T-bet$^-$) BM-$T_{SCM}$ predicts relapse. Accordingly, leukemia-specific T cells in patients prone to relapse display exhaustion markers, absent in patients maintaining long-term CR. These results highlight a wide, though reversible, immunological dysfunction in the BM of AML patients relapsing after HSCT and suggest new therapeutic opportunities for the disease.

[1] IRCCS San Raffaele Scientific Institute, Division of Immunology, Transplantation, and Infectious Diseases, Experimental Hematology Unit, via Olgettina 60, Milan 20132, Italy. [2] IRCCS San Raffaele Scientific Institute, Hematology and Hematopoietic Stem Cell Transplantation Unit, via Olgettina 60, Milan 20132, Italy. [3] IRCCS San Raffaele Scientific Institute, Division of Immunology, Transplantation and Infectious Diseases, Experimental Immunology Unit, via Olgettina 60, Milan 20132, Italy. [4] IRCCS San Raffaele Scientifc Institute, Division of Immunology, Transplantation and Infectious Disease, Unit of Immunogenetics, Leukemia Genomics and Immunobiology, via Olgettina 60, Milan 20132, Italy. [5] IRCCS San Raffaele Scientific Institute, Division of Immunology, Transplantation and Infectious Diseases, Innovative Immunotherapies Unit, via Olgettina 60, Milan 20132, Italy. [6] Hokkaido University Faculty of Medicine, Graduate School of Medicine, Department of Hematology, 5 Chome Kita 8 Jonishi, Kita, Sapporo 060-0808, Japan. [7] Leiden University Medical Center (LUMC), Department of Hematology, Albinusdreef 2, Leiden 2333, The Netherlands. [8] Department of Internal Medicine I, University Hospital Carl Gustav Carus Dresden, Technical University Dresden, Fetsherstrasse 74, Dresden 01307, Germany. [9] University of Freiburg Medical Center, Department of Hematology, Oncology and Stem Cell Transplantation, Hugstetterstr 55, Freiburg 79106, Germany. [10] Università Vita-Salute San Raffaele, via Olgettina 60, Milan 20132, Italy. [11] Present address: University of Milan, via Festa del Perdono 7, Milan 20122, Italy. These authors contributed equally: Maddalena Noviello, Francesco Manfredi. These authors jointly supervised this work: Luca Vago, Fabio Ciceri, Chiara Bonini. Correspondence and requests for materials should be addressed to C.B. (email: bonini.chiara@hsr.it)

In patients affected by high-risk hematological malignancies, such as acute myeloid leukemia (AML), allogeneic hematopoietic stem cells transplantation (HSCT) represents the most effective treatment option. Still, disease relapse and progression remain the major causes of treatment failure[1]. HSCT efficacy largely relies on the ability of donor T cells to eliminate residual tumor cells, through a phenomenon described as Graft-versus Leukemia (GvL) effect[2]. Durable immunosurveillance after HSCT likely requires long-term persistence of such leukemia-reactive T cells, possibly maintained by a stem-cell-like memory T-cell pool[3,4]. Indeed, according to the hierarchical model of T-cell differentiation[5], after antigen encounter, naive T cells differentiate into several functional subsets, including central memory ($T_{CM}$), effector memory ($T_{EM}$), and terminal effectors ($T_{EM}RA$). Memory stem T cells ($T_{SCM}$)[6] are a newly described subset that differentiate directly from naive T cells upon TCR engagement and retain the capacity of self-renewal and to hierarchically differentiate into all other memory T-cell subsets[7,8]. Clonal tracking of genetically modified T cells infused into patients affected by malignant and non-malignant diseases revealed the ability of $T_{SCM}$ to persist for decades in the host and to recapitulate the ontogeny of circulating memory T cells[9,10].

Even when immune reconstitution is preserved and maintained long-term after transplant, leukemic blasts can escape the immune response by several mechanisms[11]. At the tumor cell level, a combination of genomic instability and a Darwinian process of immunoselection may ultimately lead to a loss of tumor immunogenicity. For instance, by monitoring patients relapsing after mismatched HSCT, we described the loss of the host's mismatched HLA haplotype by leukemic cells as a relevant biological mechanism leading to tumor escape and clinical disease recurrence[12,13], particularly frequent in late relapses[14]. Alternatively, the presence of tolerogenic $T_{regs}$ or cells expressing inhibitory ligands such as PD-L1[15] may result in the loss of donor-mediated antitumor activity.

In the last years, the expression of multiple inhibitory receptors on the cell surface of antigen-experienced T cells has been associated to T-cell "exhaustion", a functional status characterized by concomitant loss of cytokines production, proliferative capacity, and lytic activity[16]. First described in chronic infections, T-cell exhaustion is considered a common and relevant phenomenon in cancer progression, as well demonstrated by the efficacy of immune checkpoint-blocking therapy, a paradigm-shifting treatment for several tumors[17]. In the setting of leukemia, a pioneering study reported the efficacy of anti-CTLA-4 blocking antibody as a treatment of post-transplantation relapse[18]. However, data on the role of immune checkpoints in the control of hematological malignancies are still limited.

In the current study, we investigated whether T-cell exhaustion is involved in the development of post-transplant leukemic relapse. To this end, we evaluated the expression of several inhibitory receptors on different bone marrow (BM) infiltrating memory CD4$^+$ and CD8$^+$ T-cell subsets in AML patients who received HSCT. We identified a PD-1$^+$ TIM-3$^+$ KLRG1$^+$ 2B4$^+$ exhaustion signature that characterizes early-differentiated CD8$^+$ BM-$T_{SCM}$ and $T_{CM}$ subsets, during disease relapse.

## Results

**Increased frequency of BM-$T_{regs}$ associates to AML relapse.** We analyzed BM and peripheral blood (PB) from 32 patients affected by AML who received HSCT from either HLA-matched (20 pts) or HLA-haploidentical (12 pts) donors. Clinical characteristics of patients are summarized in Table 1. Samples were collected at relapse (REL; median 251 days after HSCT; 16 pts) or, for patients who achieved and maintained complete remission (CR; 16 pts), at 1

year after HSCT. Samples from 11 healthy donors (HD) were used as controls. The gating strategy of the flow-cytometry analysis is reported in Supplementary Fig. 1. After transplant, T cells infiltrating the BM (BM-T cells) of patients in CR displayed an inverted CD4/CD8 ratio compared with HD ($p < 0.0001$), consistent with a more rapid CD8$^+$ T-cell reconstitution, whereas relapsing patients displayed a wider variability in CD4/CD8 ratio both in BM (Supplementary Fig. 2a) and PB (Supplementary Fig. 2b). As already reported[19], T regulatory cells ($T_{regs}$), defined by the coexpression of CD4, CD25, and FoxP3, represented a small cellular population in the bone marrow. The frequency of BM-infiltrating $T_{regs}$ was significantly higher in REL patients compared with CR patients ($p < 0.01$) and HD ($p < 0.01$) (Supplementary Fig. 2c). This difference held also when separately evaluating naive (defined as CD45RA$^+$FoxP3$^{low}$) and activated (CD45RA$^-$FoxP3$^{high}$) cells (Supplementary Fig. 2d), the two major $T_{reg}$ subsets[20]. Interestingly, this observation was confined to tumor site, since no significant differences in $T_{reg}$ frequencies were observed among patient cohorts in PB (Supplementary Fig. 2e, f).

Next, we evaluated CD4$^+$ and CD8$^+$ T-cell differentiation by assessing CD62L, CD45RA, and CD95 expression[4,21] (Supplementary Fig. 2g). In transplanted patients, we observed a lower proportion of CD4$^+$ and CD8$^+$ BM-T naive ($T_{naive}$) ($p < 0.001$) and a concomitant increase of $T_{EM}$ ($p < 0.0001$) lymphocytes than in HDs, with no significant differences between CR and REL patients (Fig. 1a). A similar T-cell distribution was observed in PB (Supplementary Fig. 2h, i). The analysis of the TCR repertoire indicated a skewed BM-T-cell clonal composition in transplanted patients, independent of clinical outcome. Analyses of TCR-α chain showed that the 10 most frequent clonotypes represent up to 33 and 39% of all detected clonotypes in CR and REL patients, respectively, and only 22% in HD; a similar T-cell skewing was observed upon TCR-β-chain sequencing (Fig. 1b). These results show that the transplant procedure is the major factor promoting the expansion of CD8$^+$ T cells, their differentiation into effectors and the skewing of the TCR repertoire toward few dominant clones, whereas leukemia relapse is associated to a high frequency of $T_{regs}$, specifically at the disease site.

**Exhausted phenotype of CD8$^+$ BM-T cells at AML relapse.** To assess the functional and exhaustion profile of BM-T cells, we analyzed the expression of costimulatory molecules and inhibitory receptors (IRs) on CD4$^+$ and CD8$^+$ lymphocytes and observed a different pattern of expression according to the level of HLA disparity between donor and host. The gating strategy of the flow-cytometry analysis is reported in Supplementary Fig. 1. Patients receiving HLA-haploidentical HSCT had lower percentages of CD27$^+$CD4$^+$ T cells and CD28$^+$CD8$^+$ T cells, compared with healthy donors, in line with the high proportion of effector T lymphocytes observed after HSCT. In this group of patients, with the exception of a slightly but significant lower proportion of T cells expressing LAG-3 in REL than in CR patients ($p < 0.05$), the profile of IR expression in BM-T cells did not correlate with clinical outcome (Fig. 1c).

After HLA-identical HSCT, the expression profile of CD4$^+$ T cells appeared again dominated by the transplant procedure, and distinct from that of HD, with a higher percentage of CD4$^+$ T-cells expressing ICOS and inhibitory receptors in transplanted patients than in HD (Fig. 1d). In contrast, the exhaustion profile of CD8$^+$ BM-T cells varied significantly, according to the clinical outcome. In fact, we observed a higher proportion of CD8$^+$ BM-T cells expressing CTLA-4 (median 29.2%), PD-1 (44.1%), and TIM-3 (10.6%) inhibitory molecules in REL patients than in CR patients (CTLA-4 19.4%, PD-1 24.0%, TIM-3 2.8%; $p < 0.05$) (Fig. 1d). Interestingly, some, but not all phenotypic

**Table 1 Patients' characteristics at long-term follow-up**

| | HLA-matched HSCT long-term complete remission (CR) | HLA-matched HSCT relapse (REL) | Haploidentical HSCT long-term complete remission (CR) | Haploidentical HSCT relapse (REL) |
|---|---|---|---|---|
| Number of patients, n | 10 | 10 | 6 | 6 |
| Diagnosis n (%) | | | | |
| AML | 8 (80%) | 9 (90%) | 6 (100%) | 5 (83%) |
| MDS | 2 (20%) | 1 (10%) | 0 (0%) | 1 (17%) |
| Donor type, n (%) | | | | |
| HLA-matched sibling | 5 (50%) | 5 (50%) | 0 (0%) | 0 (0%) |
| HLA-matched MUD (9–10/10) | 5 (50%) | 5 (50%) | 0 (0%) | 0 (0%) |
| HLA-haploidentical | 0 (0%) | 0 (0%) | 6 (100%) | 6 (100%) |
| CMV serostatus donor/recipient, n (%) | | | | |
| pos/pos | 5 (50%) | 7 (70%) | 4 (67%) | 4 (67%) |
| pos/neg | 0 (0%) | 0 (0%) | 0 (0%) | 0 (0%) |
| neg/pos | 5 (50%) | 2 (20%) | 2 (33%) | 2 (33%) |
| neg/neg | 0 (0%) | 1 (10%) | 0 (0%) | 0 (0%) |
| Disease status at transplant, n (%) | | | | |
| Complete remission | 8 (80%) | 8 (80%) | 3 (50%) | 1 (17%) |
| Presence of disease | 2 (20%) | 2 (20%) | 3 (50%) | 5 (83%) |
| Conditioning regimen, n (%) | | | | |
| Reduced-intensity | 2 (20%) | 3 (30%) | 0 (0%) | 0 (0%) |
| Myeloablative, treosulfan-based | 6 (60%) | 4 (40%) | 6 (100%) | 6 (100%) |
| Myeloablative, other | 2 (20%) | 3 (30%) | 0 (0%) | 0 (0%) |
| In vivo T-cell depletion, n (%) | | | | |
| None | 6 (60%) | 5 (50%) | 0 (0%) | 0 (0%) |
| ATG | 5 (50%) | 1 (10%) | 3 (50%) | 0 (0%) |
| PT-Cy | 0 (0%) | 0 (0%) | 3 (50%) | 6 (100%) |
| ATG/PT-Cy | 1 (10%) | 4 (40%) | 0 (0%) | 0 (0%) |
| GvHD prophylaxis, n (%) | | | | |
| CSA-based | 8 (80%) | 5 (50%) | 0 (0%) | 0 (0%) |
| Sirolimus-based | 1 (10%) | 2 (20%) | 6 (100%) | 6 (100%) |
| Other | 1 (10%) | 3 (30%) | 0 (0%) | 0 (0%) |
| Clinically relevant post-HSCT CMV reactivation*, n (%) | | | | |
| Yes | 4 (40%) | 4 (40%) | 1 (17%) | 1 (17%) |
| No | 6 (60%) | 6 (60%) | 5 (83%) | 5 (83%) |
| Acute GvHD incidence, n (%) | | | | |
| Grade 0-I | 8 (80%) | 9 (90%) | 6 (100%) | 6 (100%) |
| Grade II-IV | 2 (20%) | 1 (10%) | 0 (0%) | 0 (0%) |
| Chronic GvHD incidence, n (%) | | | | |
| None | 9 (90%) | 8 (80%) | 3 (50%) | 6 (100%) |
| Mild | 1 (10%) | 1 (10%) | 0 (0%) | 0 (0%) |
| Moderate | 0 (0%) | 0 (0%) | 0 (0%) | 0 (0%) |
| Severe | 0 (0%) | 1 (10%) | 3 (50%) | 0 (0%) |
| Under treatment for GvHD at the time of sampling, n (%) | | | | |
| None | 10 (100%) | 10 (100%) | 6 (100%) | 6 (100%) |
| Yes | 0 (0%) | 0 (0%) | 0 (0%) | 0 (0%) |
| Time point sampling (days after HSCT), median (range) | 425 (270–1438) | 589 (229–1299) | 239 (137–364) | 204 (149–237) |

*AML* acute myeloid leukemia, *MDS* myelodysplasia, *ATG* anti-thymocyte antibodies, *PT-Cy* post-transplant cyclophosphamide, *GvHD* graft-versus-host disease, *HSCT* haematopoietic stem cell transplant, *CMV* cytomegalovirus
*Clinically relevant CMV reactivation is defined as a CMV reactivation requiring systemic preemptive antiviral treatment in accordance to istitutional guidelines (DNAemia > 1000 cp/ml, measured on plasma)

perturbations detected in BM-T cells were also observed in the peripheral blood (Supplementary Fig. 3a, b).

Of notice, at relapse, leukemic blasts expressed CD80 and CD86, ligands for both CD28 and CTLA-4, PD-L1 (ligand of PD-1), CD48 (ligand of 2B4), and Galectin-9 (ligand of TIM-3, Fig. 1e), suggesting that the inhibitory receptors expressed by BM-T cells could be triggered directly by the relapsing blasts. Interestingly, a positive correlation was observed between the proportion of TIM-3+ CD8+ T cells and the frequencies of Galectin-9+ AML blasts in HLA-matched transplanted patients (Supplementary Fig. 3c).

Overall, our results reveal a dominant effect of the transplant procedure on the BM-T-cell functional profile, lasting 1 year after

transplant. In addition, in patients relapsing after HLA-matched HSCT, we observed that PD-1+CTLA-4+TIM-3+ exhausted CD8+ T cells accumulate in the bone marrow. The pattern of inhibitory ligand expression on AML blasts at relapse implies an active immunosuppressive process, underscoring the possible relevance of these IRs in AML progression. According to these observations, we further investigated the functional profile of BM-T cells in patients receiving HLA-matched HSCT.

**BH-SNE algorithm segregates IR+ relapse-specific clusters.** To validate our findings by an unsupervised approach and identify putative cellular populations associated to clinical outcome, we

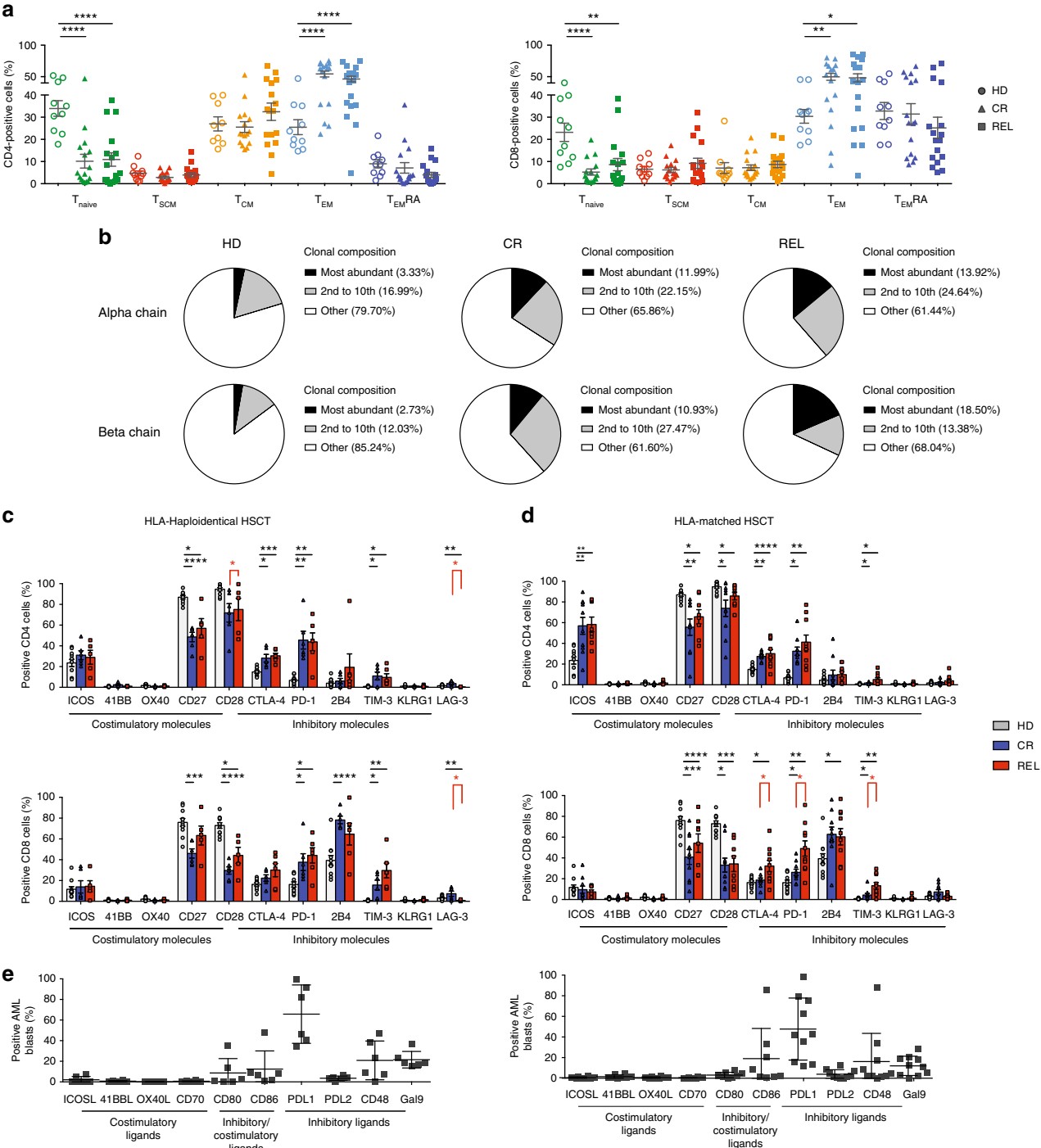

**Fig. 1** T-cell subset phenotype in the bone marrow of HSCT patients differs according to the clinical outcome and transplant type. **a** Proportion of naive (CD45RA$^+$CD62L$^+$CD95$^-$, T$_{naive}$), memory stem (CD45RA$^+$CD62L$^+$CD95$^+$, T$_{SCM}$), central memory (CD45RA$^-$CD62L$^+$, T$_{CM}$), effector memory (CD45RA$^-$CD62L$^-$, T$_{EM}$) T cells and terminal effectors (CD45RA$^+$CD62L$^-$, T$_{EM}$RA) over the total CD4$^+$ and CD8$^+$ BM-T-cell subsets in healthy donors (HD, $N = 11$), in patients who achieved long-term complete remission (CR, $N = 16$) and in patients who experienced relapse (REL, $N = 16$) after HSCT. **b** Pie charts depicting the relative abundance of T-cell clones in each study group (HD, $N = 3$; CR, $N = 3$; REL, $N = 3$) obtained after CDR3 sequencing of both TCR-α and TCR-β chains. **c**, **d** Percentages of BM-infiltrating CD4$^+$ and CD8$^+$ T cells positive for costimulatory or inhibitory receptors in HLA-haploidentical ($N = 12$, **c**) or HLA-matched ($N = 20$, **d**) transplant settings; bone marrow samples from healthy donors were used as controls (HD, $N = 11$). **e** Percentages of leukemic blasts at relapse expressing costimulatory or inhibitory ligands in patients receiving HLA-haploidentical (left) or HLA-matched (right) transplants. All the inhibitory receptors have been visualized by means of cell surface staining apart for CTLA-4, visualized after cell fixation and permeabilization. Individual patient data points, mean, and SEM are shown. Statistically significant differences between CR and REL groups are highlighted in red, the differences between patients' groups and HD in black. *$p < 0.05$; **$p < 0.01$; ***$p < 0.001$; ****$p < 0.0001$; nonparametric unpaired two-sided $T$ test

analyzed our data with BH-SNE (Barnes-Hut Stochastic Neighbor Embedding[22]), a dimensionality reduction algorithm that attributes two new variables (BH-SNE1 and BH-SNE2) to each event (i.e., the single cell in flow cytometry), considering as discriminants all the fluorochromes at a time.

HLA-matched patients ($N = 20$) and healthy donors ($N = 10$) bone marrow samples were analyzed by BH-SNE after unsupervised and stochastic data downscaling, in which ~7500 events among $CD3^+$ lymphocytes were chosen from each file (Fig. 2a). The two newly calculated variables BH-SNE1 and BH-SNE2 allowed data to be visualized in a single two-dimensional plot. If divided according to three experimental groups, HD, CR, and REL, the events clustered in different areas. When considering a region of high density for each experimental group (black rectangles in Fig. 2a), higher mean fluorescence intensities (MFI) for inhibitory receptors in case of REL were observed.

To better identify discrete spatial regions of the BH-SNE plot specifically associated to each experimental group, we deployed the clustering algorithm K-means (Fig. 2b). First, we calculated K-means on BH-SNE1 and BH-SNE2 variables, dividing the plot in 200 meta-clusters; then, we attributed meta-clusters to either HD, CR or REL when more than 55% of the events were specific for one experimental group. Two series of BH-SNE analysis for each sample were performed, since the markers of interest were divided into two distinct flow-cytometry panels (panel 1 and panel 2, Fig. 2c). In total, 32 and 34 meta-clusters were attributed to HD in panel 1 and in panel 2, respectively; 9 and 16 meta-clusters were attributed to CR while 15 and 17 to REL. We calculated the relative fluorescence intensity (RFI) for each marker of each meta-cluster in panels 1 and 2 and observed a higher RFI in case of relapse for PD-1, KLRG1 and TIM-3 when compared with HD and CR (Fig. 2d). The analysis with BH-SNE algorithm also revealed that most (81.3%) HD meta-clusters did not express IRs, while only a fraction (17.7%) of them expressed only one IR. In the CR group, the majority (64.0%) of meta-clusters was negative for IRs, while in REL patients only one-third was IR negative and 67.8% expressed one or more IRs (Fig. 2e). In conclusion, this unsupervised approach revealed the presence, in relapsing patients, of distinct T-cell populations characterized by high expression of multiple inhibitory receptors.

**Early-differentiated $CD8^+$ BM-T cells at relapse are exhausted**. Since most differences in IR profiles observed in REL versus CR patients clustered in $CD8^+$ BM-T cells, we focused on this T-cell subset and investigated how the exhaustion profile was modulated at different stages of memory $CD8^+$ T-cell differentiation in the BM of patients after HLA-matched HSCT. The gating strategy of the flow-cytometry analysis is reported in Supplementary Fig. 1. In HD (Fig. 3a, b), the expression of inhibitory receptors 2B4 and PD-1 was confined to $T_{EM}$ (mean 2B4: 69.2%; PD-1: 25.7%) and terminal effectors $T_{EM}RA$ (2B4 69.8% and PD-1 11.0%), while CTLA-4 expression was more equally distributed in all differentiation stages, and TIM-3, KLRG1 and LAG-3 were barely detectable. In CR patients, $CD8^+$ BM-T cells recapitulated almost completely the pattern observed in HD. At relapse, $T_{EM}$ and $T_{EM}RA$ showed an inhibitory receptor profile similar to that of CR patients, with the exception of PD-1, more homogeneously expressed (46.4% and 29.6%, respectively in $T_{EM}$ and $T_{EM}RA$). Strikingly, however, in REL patients, a high proportion of early-differentiated $CD8^+$ T cells displayed an exhausted phenotype. A higher frequency of $T_{CM}$ expressed 2B4 (mean 54.5%), PD-1 (33.8%), and TIM-3 (14.6%) in REL patients compared with HDs (2B4 12.0%, $p < 0.0001$; PD-1 9.8%, $p < 0.01$; TIM-3 1.9%, $p < 0.01$). Furthermore, the proportion of TIM-3$^+$ $T_{CM}$ cells was higher in REL than in CR patients (3.6%, $p < 0.01$). These

differences were even more pronounced in BM-$T_{SCM}$ cells, a higher percentage of which expressed PD-1 (30.6%), TIM-3 (15.1%), and KLRG1 (14.7%) in REL patients than in HDs (PD-1 3.8%, TIM-3 1.1%, KLRG1 1.6%; $p < 0.001$) or CR patients (PD-1 7.6%, $p < 0.01$; TIM-3 3.7%, $p < 0.001$; KLRG1 1.4%, $p < 0.001$). Finally, a higher proportion of BM-$T_{SCM}$ cells from REL patients expressed 2B4 (mean 51.0%) compared with HDs (2B4 23.0%, $p < 0.05$).

Unsupervised clustering analysis summarized in the heatmap of Fig. 3c shows (left to right) an enrichment of BM-T cells expressing IRs correlating with progressive T-cell differentiation and unfavorable clinical outcome. Interestingly, the right part of the heatmap contains late differentiated T cells of all patient cohorts and early-differentiated T cells only belonging to relapsing patients, indicating that this T-cell subset is assimilated to more differentiated (and exhausted) effectors when relapse occurs. Altogether, these results show that early-differentiated $CD8^+$ BM-T cells in patients who relapse after HLA-matched HSCT display an exhausted phenotype.

**BM-T cells at AML relapse are functionally impaired**. Polyfunctionality is a relevant measure of T-cell fitness[23]. We thus evaluated the function of BM-T cells by analyzing the production of IL-2, IFN-γ, and TNF-α cytokines and the expression of CD107a upon activation with a polyclonal stimulus. The proportions of both early $CD62L^+$ (Fig. 4a) and late $CD62L^-$ (Fig. 4b) differentiated BM-T cells simultaneously displaying four effector functions were lower in REL patients compared with HD and CR. In relapsing patients, T-cell hyporesponsiveness appeared largely due to reduced degranulation, since both early- and late-differentiated T cells showed defects in CD107a expression, both when measured alone or in combination with cytokine production (Fig. 4a, b). Altogether these data indicate that at the primary disease site both early- and late-differentiated T cells manifest a partial functional impairment, in accordance with the inhibitory receptor profile in relapsing patients.

**IR$^+$ BM-T cells at relapse display leukemia specificities**. Exhaustion of tumor-infiltrating lymphocytes is currently ascribed to several mechanisms, including inhibitory signals and chronic antigen exposure promoted by cancer cells and tumor microenvironment[24]. It can thus be assumed that exhausted lymphocytes at the tumor site are enriched in tumor-specific T cells. To verify this hypothesis, BM-T cells from six REL patients were sorted in IR$^+$ cells, based on the expression of at least one inhibitory receptor, and IR$^-$ cells (Fig. 5a). The comparative analyses of the TCR repertoire showed a higher abundance of the 10 strongest clonotypes in IR$^+$ than in IR$^-$ lymphocytes, confirmed with both TCR-α and β-chain sequences (Fig. 5b), suggesting a more restricted T-cell repertoire in IR$^+$ cells. Sorted IR$^+$ and IR$^-$ BM-T cells expanded in vitro with a similar kinetic and magnitude upon rapid expansion protocol and high doses of IL-2[25] (Fig. 5c), indicating that the exhausted phenotype can be reverted in culture, as already reported[26]. Upon in vitro culture, IR$^+$ cells recovered the ability to degranulate and produce IFN-γ and TNF-α (Fig. 5d, e); however, expanded cells differentiate in late effector T cells and, accordingly, did not fully recover IL-2 production capacity. Most interestingly, despite no enrichment for leukemia-specific T cells was performed in vitro, a higher proportion of IR$^+$ lymphocytes recognized autologous leukemic blasts compared with IR$^-$ cells in vitro, as demonstrated by granzyme A and B production (IR$^+$ and IR$^-$: 762.6 pg/mL, vs. 0 pg/mL; $p < 0.001$; Fig. 5f) and cytotoxicity (elimination index: 62.2% for IR$^+$ vs. 43.5% for IR$^-$ at 100:1 E:T ratio; $p < 0.05$; Fig. 5g). Taken together, these results indicate that exhausted

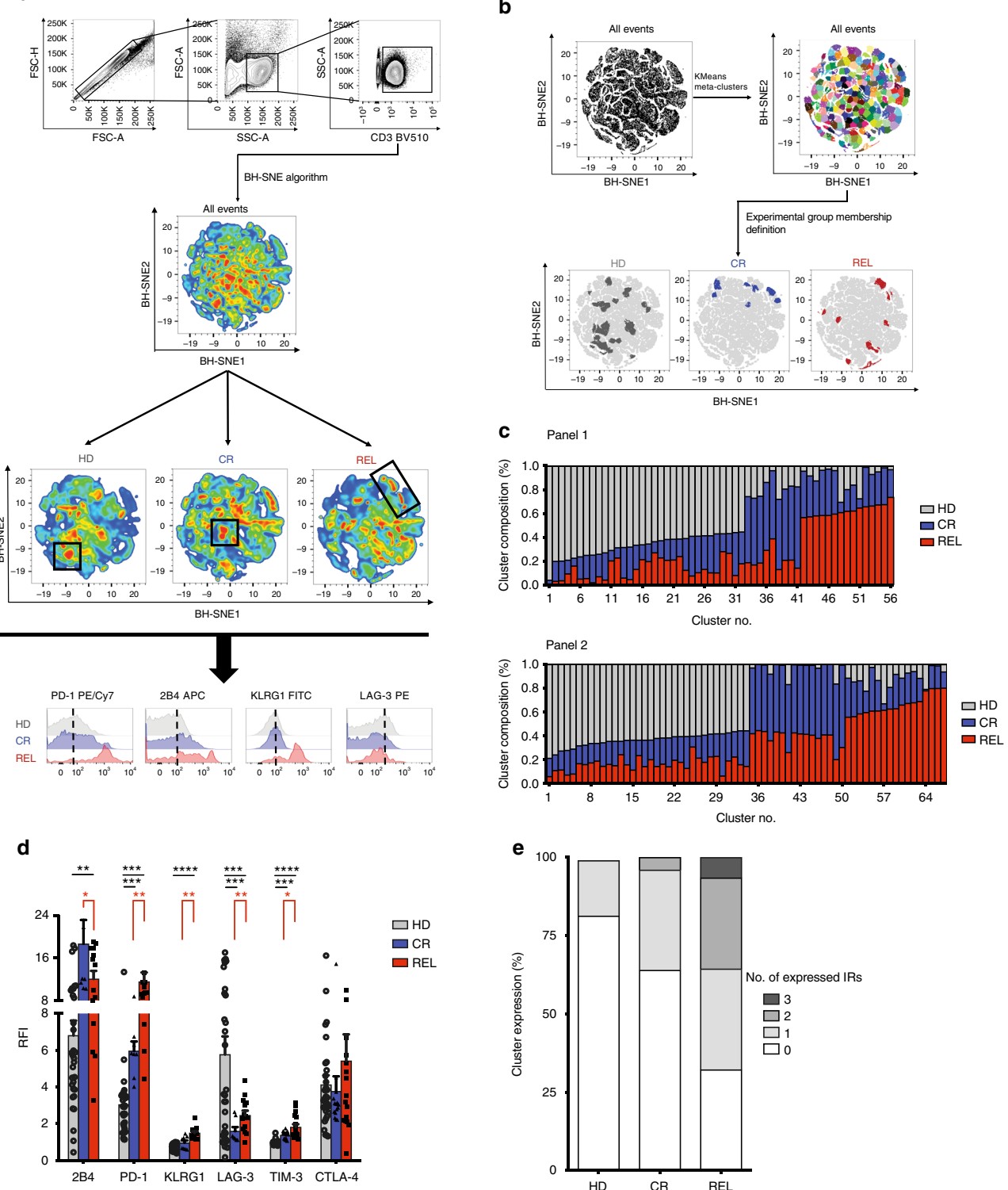

**Fig. 2** BH-SNE and K-means tandem application evaluates IR coexpression. **a** A median of 7500 CD3$^+$ lymphocytes from bone marrow samples were studied with BH-SNE algorithm (top) and plotted according to the calculated variables BH-SNE1 and BH-SNE2 (mid); the events were then split into three density plots according to the study group they belong to (healthy donors, $N = 10$ HD; patients achieving long-term remission after transplant, CR $N = 10$; and patients experiencing post-transplant relapse, REL $N = 10$); representative histograms of PD-1, 2B4, KLRG1, and LAG-3 from selected areas are reported (bottom). **b** K-means algorithm was applied to BH-SNE1 and BH-SNE2 variables, and a fraction of the identified meta-clusters were attributed to each experimental group (HD, CR, and REL). **c** Composition of meta-clusters ascribed to a specific experimental group (variables analyzed in panel 1 and panel 2 are detailed in the Methods section). **d**, **e** HD-, CR-, or REL-specific meta-clusters were described in terms of both relative fluorescence intensity (RFI, **d**) and multiple expression (**e**) of inhibitory receptors. Individual patients datapoint, means, and SEM are shown. Statistically significant differences between CR and REL groups are highlighted in red and the differences between patients' groups and HD in black. *$p < 0.05$; **$p < 0.01$; ***$p < 0.001$; ****$p < 0.0001$; nonparametric unpaired two-sided $T$ test

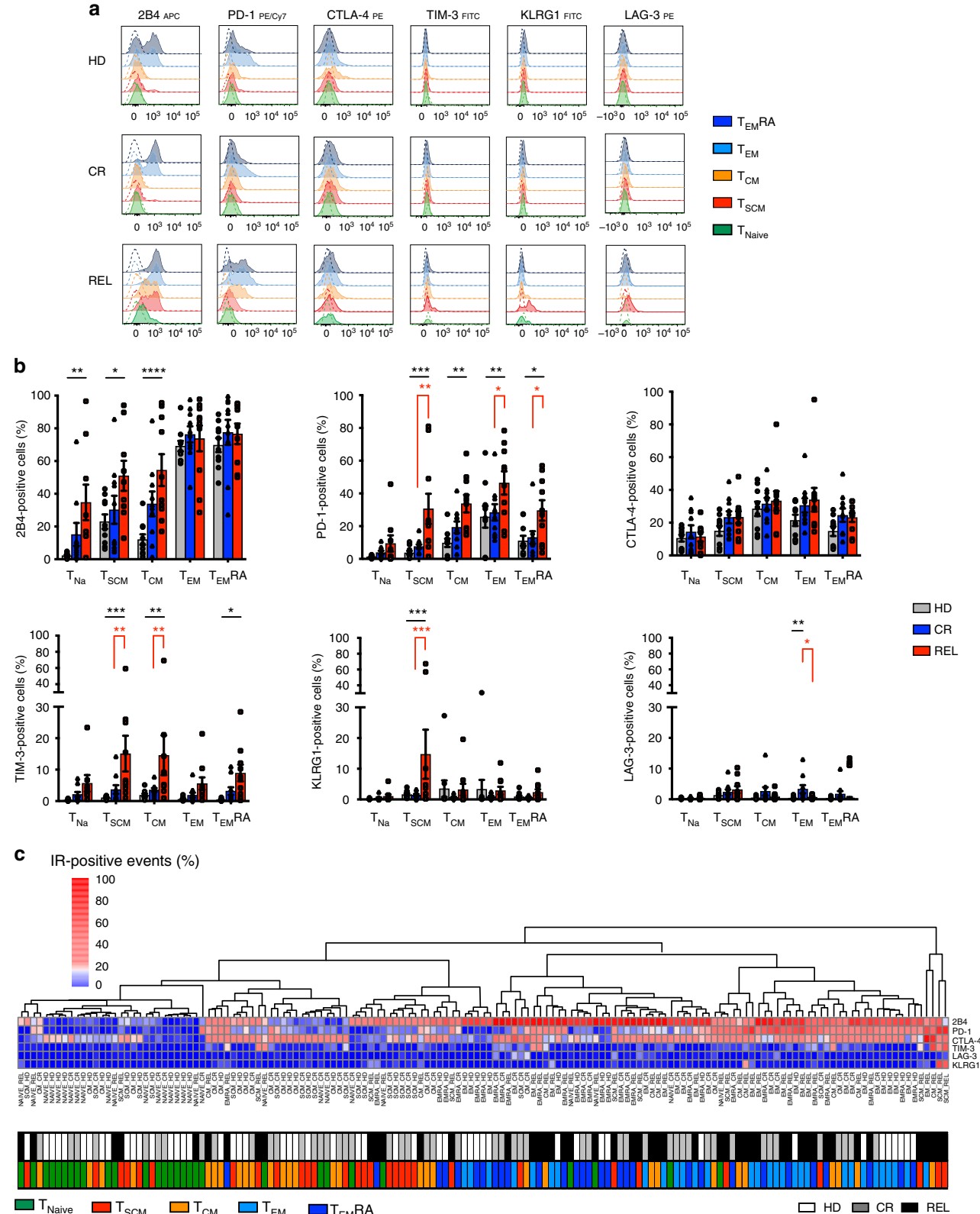

T cells at the tumor site preferentially harbor specificities directed against leukemic blasts.

**Tumor-specific CD8+ BM-T cells are exhausted early after HSCT.** Having observed that BM-infiltrating T cells of HLA-matched patients are exhausted at the time of disease recurrence,

we hypothesized that this dysfunctional profile could be detected also before relapse.

We retrospectively analyzed T cells harvested from the BM of 57 patients at early time points (median 68 days) after HLA-matched HSCT. Patients characteristics are detailed in Table 2, and gating strategies for flow-cytometry data are summarized in

**Fig. 3** Exhausted CD8$^+$ T$_{SCM}$ and T$_{CM}$ infiltrate the bone marrow of relapsing patients. **a** Representative histograms showing the inhibitory receptor expression profile in CD8$^+$ naive (CD45RA$^+$CD62L$^+$CD95$^-$, T$_{naive}$), memory stem (CD45RA$^+$CD62L$^+$CD95$^+$, T$_{SCM}$), central memory (CD45RA$^-$CD62L$^+$, T$_{CM}$), effector memory (CD45RA$^-$CD62L$^-$, T$_{EM}$) T cells, and terminal effectors (CD45RA$^+$CD62L$^-$, T$_{EM}$RA) in healthy donors (HD, $N = 10$), in HLA-matched patients maintaining complete remission (CR, $N = 10$) or relapsing after transplant (REL, $N = 10$). Fluorescence-minus one (FMO) negative controls (dashed lines) are shown for each subpopulation. **b** Relative proportion of IR-expressing CD8$^+$ T cells in each memory subpopulation from the different experimental groups (HD, CR, and REL). **c** Heatmap representing percentage of positive cells for all samples classified according to the memory differentiation status ($N = 5$ inputs per samples, $N = 150$ total inputs); every input was associated with two colored tags, one referring to the memory status (T$_{Na}$ green, T$_{CM}$ orange, T$_{SCM}$ red, T$_{EM}$ light blue, T$_{EM}$RA dark blue) and one to the experimental group (HD white, CR gray, REL black). All the inhibitory receptors have been visualized by means of cell surface staining apart for CTLA-4, visualized after cell fixation and permeabilization. Individual patient data points, means, and SEM are shown. Statistically significant differences between CR and REL groups are highlighted in red and the differences between patients' groups and HD in black. *$p < 0.05$; **$p < 0.01$; ***$p < 0.001$; ****$p < 0.0001$; two-way ANOVA with no matching coupled with Sidak multiple correction test

Supplementary Fig. 4. We detected a low CD4/CD8 T-cell ratio (Supplementary Fig. 5a) with a percentage of naive CD4$^+$ and CD8$^+$ BM-T cells lower in transplanted patients than in HD, independently of the clinical outcome, while effectors appeared enriched (Supplementary Fig. 5b, c). The expression of IRs was high and neither did not differ according to the clinical outcome, when studied on total CD8$^+$ BM-T cells, nor on T-cell memory subpopulations (Supplementary Fig. 5d, e) indicating that the transplant procedure dominates and shapes the CD8$^+$ T-cell profile early after transplant. We hypothesize that in this early phase, in which alloreactivity is dominant, the expression of inhibitory receptors might reflect the largely activated status of BM-T cells. According to this hypothesis, a large proportion of CD8$^+$ BM-T cells, at early time points expressed the HLA-DR activation marker (Supplementary Fig. 5d). In such a highly activated immunological setting, we attempted to identify putative terminally exhausted T cells among total CD8$^+$ BM-T cells, defined based on the expression of PD1, Eomes and T-bet[27]. As shown in Fig. 6a, we found a significant enrichment in PD-1$^+$ Eomes$^+$ T-bet$^-$ T$_{SCM}$ cells in the BM of REL patients compared with the CR ones, suggesting that a small fraction of severely exhausted memory T$_{SCM}$ preferentially accumulates early after transplant in patients prone to relapse. To verify our results with an unsupervised approach, sensitive to small cellular populations, we applied the BH-SNE algorithm, and observed that 6 (54%) out of 11 and 11 (84%) of the 13 clusters specific for CR and REL, respectively, were negative for the expression of HLA-DR. The analysis of these HLA-DR negative clusters revealed higher expression of TIM3, 2B4, LAG3, GITR, and KLRG1 in REL than in CR (Fig. 6b). Overall, this analysis indicates the presence of a small population of not activated (HLA-DR$^-$) BM-T cells that differentially express IRs in patients prone to relapse versus patients destined to long-term CR. To verify whether dysfunctional T cells are enriched in tumor-specificities at this early time point, we took advantage of MHC dextramers to identify tumor-specific (WT1, PRAME, or EZH2) and control viral-specific (cytomegalovirus-CMV) CD8$^+$ T cells from patients. We observed that viral-specific T cells were IR$^-$ independently of the ultimate clinical outcome. Furthermore, also most tumor-specific T cells retrieved from CR patients did not express IRs. By contrast, a large number of tumor-specific T cells present in the REL group expressed IRs (Fig. 6c). In particular, >50% of tumor-specific T cells in the REL group expressed at least one IR, while 29% of these cells expressed four IRs, consistent with a profound functional exhaustion (Fig. 6d). Collectively, these data strongly suggest that antitumor T-cell reactivity is already impaired early after transplant in the patients who subsequently relapse.

## Discussion
IRs modulate T-cell responsiveness and act as negative feedback mechanisms limiting T-cell activation during antigen exposure.

Recent findings show that tumors often hijack IRs to acquire protection from immune attack[24]. In the present work, we show that CD8$^+$ BM-T cells exhibit a unique exhaustion signature in patients relapsing after HLA-matched transplants and that such signature defines antitumor T cells in the setting of disease relapse. While in previous studies IR expression was assessed on circulating T cells and after transplant[28–30], we concentrated our efforts mainly on T cells infiltrating the bone marrow, the primary site of the disease, based on the concept that cancer cells and their microenvironment play a critical role in modulating T-cell exhaustion[31].

In this study, we focused on patients who relapsed after a phase of clinical remission lasting a median of 1 year. This choice had several reasons. First, in the early time points after HSCT, the pro-inflammatory milieu may alter the T-cell profile independently of disease recurrence, for example resulting in high expression of PD-1 on T lymphocytes in patients with aGvHD or CMV reactivation[32]. Second, leukemia-resistance mechanisms acquired against the immunological pressure imposed by HSCT might require time to fully unfold. Indeed, we reported that after HLA-haploidentical HSCT, HLA-loss occurs later than other forms of relapse[14]. To discriminate between the effects of the transplant procedure and those linked to the disease, the results obtained from relapsing patients were compared with those obtained both in healthy donors and in patients who maintained a long-term status of complete remission. We balanced the experimental groups with respect to relevant clinical variables, such as donor source, conditioning regimen, CMV serostatus, and GvHD incidence and evaluated patients in the absence of active inflammation, e.g., infections or GvHD.

We initially investigated the distribution of T-cell subsets both in PB and BM of transplanted patients at early and late time points. In the context of T repleted haploidentical HSCT, we and others recently showed an early wave of T$_{SCM}$ cell accumulation in the peripheral blood within the first 30 days after HSCT[7,8]. Here, we focused on the time of relapse (1 year, for non-relapsing patients) showing that the proportion of T$_{SCM}$ cells, similarly to all other memory T-cell subsets, is not affected by clinical outcome. Our data suggest that, also at late time points the transplant procedure dominates in instructing T-cell differentiation by eroding the naive T-cell compartment and promoting T$_{EM}$ accumulation in the BM, in all transplant settings, an effect likely due to the continuous exposure to viral and/or allo-antigens after HSCT. Interestingly, we observed an increased frequency of T$_{regs}$ in the BM of REL patients compared with CR. The T$_{reg}$ accumulation has been linked to T-cell exhaustion in recent studies on chronic viral infections[33] and on relapsed/refractory lymphoma patients[34]. However, the phenotypic portrait of exhausted T cells appears more complex and driven by several IRs[16,35]. We screened T cells in the BM of HSCT patients for the expression of six different IRs, i.e., PD-1, CTLA-4, TIM-3, LAG-3, 2B4, and

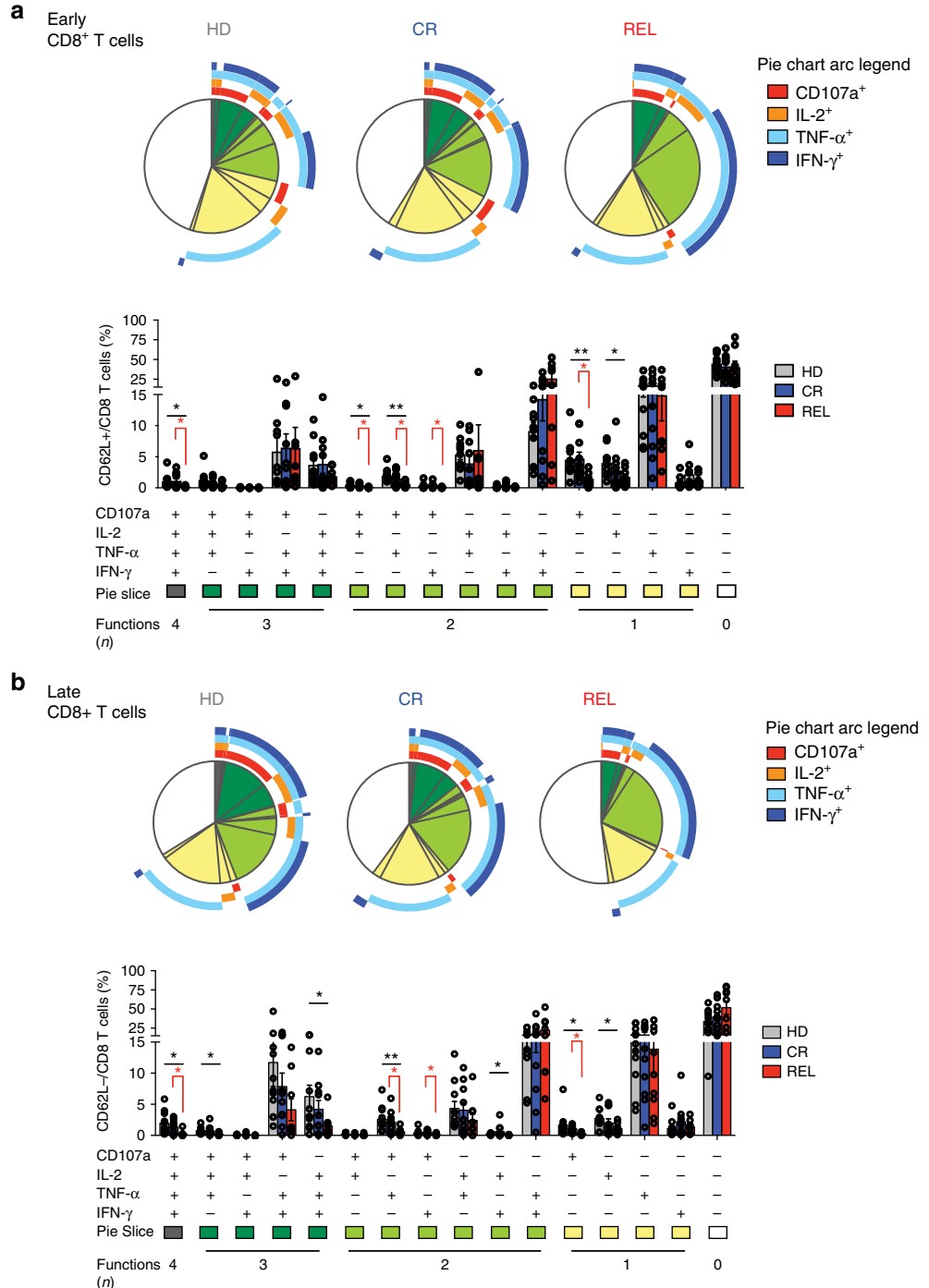

**Fig. 4** CD8[+] BM-T cells from relapsing patients display defective polyfunctionality. Cytokine production (IL-2, TNF-α, IFN-γ) and degranulation capacity (CD107a expression) of CD8[+] T cells upon PMA/Ionomycin stimulation were evaluated in bone marrow samples. **a**, **b** Pie charts with arcs (upper part) and bar chart (lower part) showing the relative proportion of CD8[+] T cells performing four, three, or two multiple effector functions or one or no effector functions in early-differentiated (CD62L[+], **a**) and late differentiated (CD62L[−], **b**) subsets. Healthy donors (HD, N = 10), HLA-matched patients maintaining complete remission (CR, N = 10) or relapsing after transplant (REL, N = 10) were evaluated. Arcs represent which function was exerted, alone or in combination with the others (CD107a, IL-2, TNF-α, and IFN-γ). Individual patient data points, means, and SEM are shown. Statistically significant differences between CR and REL groups are highlighted in red and the differences between patients' groups and HD in black. *p < 0.05; **p < 0.01; ***p < 0.001; ****p < 0.0001, nonparametric unpaired two-sided T test

KLRG1. When subjects were segregated by transplant type, we found that in the HLA-matched HSCT context, the proportion of CD8[+] T cells infiltrating the BM of REL patients expressing PD-1, CTLA-4, and/or TIM-3 was higher than in patients in CR. This was not the case in the haploidentical setting, and this effect, imputable to donor type, could be explained by the higher degree of HLA-mismatch between donor and host that could exacerbate the pro-inflammatory milieu in HLA-haploidentical patients,

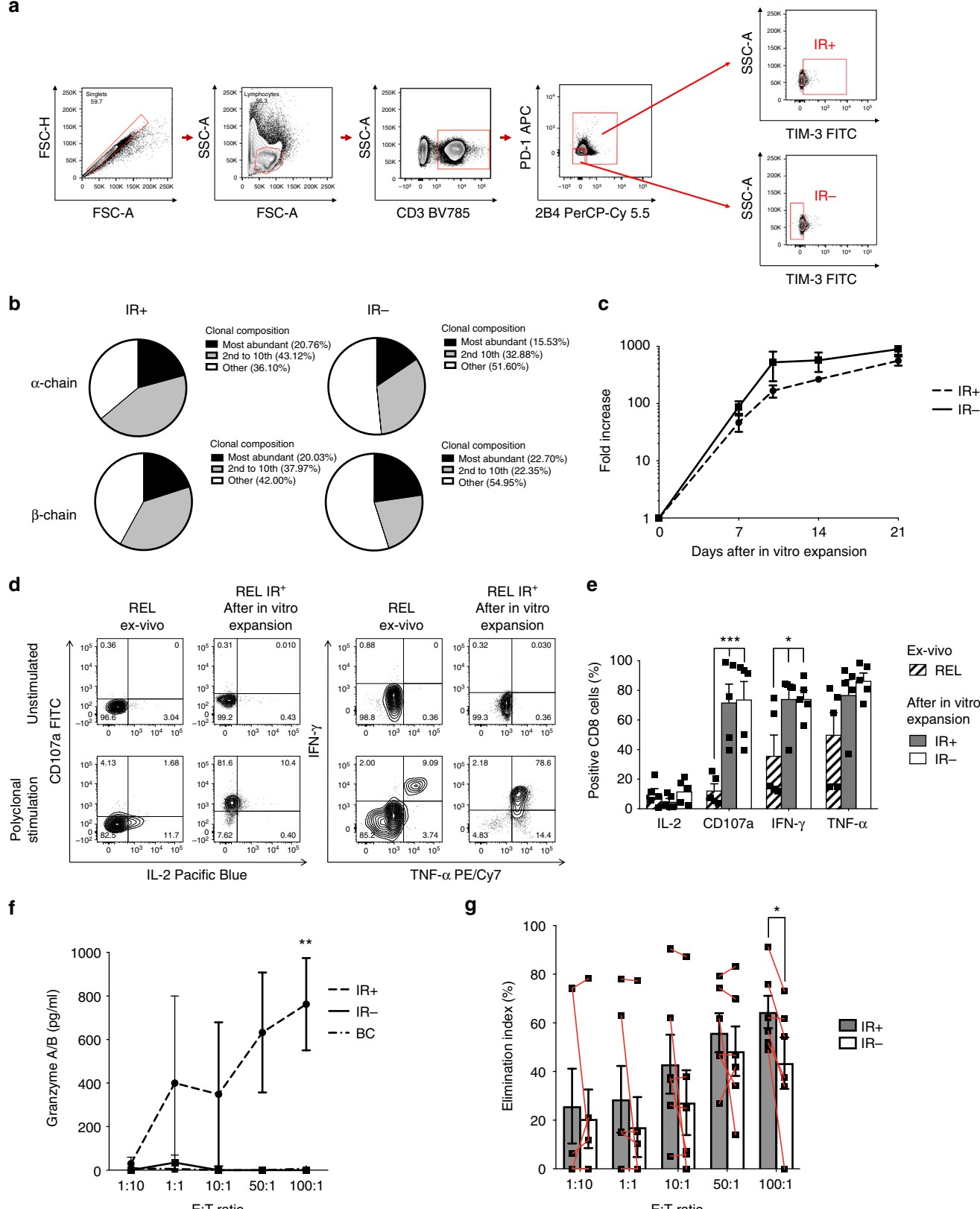

independently of the occurrence of relapse. We cannot rule-out, however, a time-dependent effect, since post-transplant samples were harvested earlier in haplo-recipients than in HLA-identical HSCT.

Given the high number of markers screened, and with the aim of analyzing them simultaneously, we implemented the BH-SNE algorithm to allow low-dimensional embedding of complex flow-cytometry data[36]. In HLA-matched patients, the unsupervised

**Fig. 5** The functional exhaustion of IR$^+$ BM-T cells at relapse is reverted in vitro. BM-infiltrating CD8$^+$ T cells from $N = 6$ patients experiencing relapse were isolated according to the expression of IRs and expanded in vitro. **a** Gating strategy for the definition of IR$^+$ (expression of at least one IR among TIM-3, PD-1, and 2B4) and IR$^-$ (TIM-3$^-$ PD-1$^-$ 2B4$^-$) T cells. **b** Pie charts depicting the relative abundance of T-cell clones in IR$^+$ ($N = 3$) and IR$^-$ ($N = 2$) cells obtained after CDR3 sequencing of both TCR-α and TCR-β chains. **c** Expansion rate of IR$^+$ and IR$^-$ T cells after in vitro polyclonal stimulation with OKT3 in presence of high doses of IL-2. **d**, **e** Representative plots (**d**) and bar graph (**e**) showing T-cell effector functions measured on ex vivo samples (dashed bars) or on in vitro expanded IR$^+$ (black bars) and IR$^-$ (white bars) T cells after short-term polyclonal stimulation (PMA/Ionomycin); two-way ANOVA with no matching coupled with Sidak multiple corrections test was used for statistical analyses. **f**, **g** IR$^+$ and IR$^-$ T cells, expanded in vitro upon polyclonal stimulation and in the absence of any procedure to enrich for leukemia-specific T cells, were co-coltured with matched AML blasts at different effector-to-target (E:T) ratios; at day 1, Granzyme A and B production was quantified (**f**), and at day 3 the elimination index (**g**) was calculated; two-way ANOVA with no matching coupled with Sidak multiple corrections test and nonparametric paired two-sided T test were used for statistical analyses, respectively. Means and SEM are shown. *$p < 0.05$; **$p < 0.01$; ***$p < 0.001$

BH-SNE analysis identified clusters of BM-T lymphocytes associated to relapse characterized by high levels of IRs and coexpression of multiple IRs, supporting the hypothesis that, at relapse, a large proportion of T cells are exhausted.

None of the IRs identified at the time of relapse was unequivocally associated to the clinical outcome when looking at the BM at early time points, before relapse. We hypothesized that this early phase after HSCT may be dominated by alloreactivity and that the expression of inhibitory receptors might reflect the largely activated status of BM-T cells, which masks the detection of exhausted cells. To better discriminate exhaustion from T-cell activation, a status that indeed includes IR expression on T cells, we took advantage of the BH-SNE algorithm and we observed that coexpression of multiple IRs on T cells lacking the HLA-DR activation marker, defines an exhausted T-cell population significantly enriched in patients prone to relapse.

The expression profile of IRs is often associated with the progressive T-cell differentiation status[37,38]. A quite remarkable finding of the present study is the observation that early-differentiated $T_{SCM}$ and $T_{CM}$ at relapse display an exhaustion signature suggesting a pervasive status of T-cell dysfunction associated with the failure to control the disease. A key question is what is the biological role of inhibited $T_{SCM}/T_{CM}$, whether they contribute to the development of leukemia relapse and whether they are responsible to the maintenance of the pool of exhausted effectors. Interestingly, a small population of PD-1$^+$ Eomes$^+$ T-bet$^-$ severely exhausted T cells, displaying a $T_{SCM}$ phenotype, were found enriched in the bone marrow of REL patients well before relapse occurred, and the coexpression of multiple IRs on BM T cells, early after HSCT, was highly associated with the risk of relapse. These results suggest that early-differentiated dysfunctional T cells may contribute to the establishment of an immunosuppressive tumor microenvironment, ultimately facilitating relapse, in accordance to the observations reported in the setting of chronic viral infections[27]. The generation of such exhausted T cells at an early time point might be the result of a continuous aberrant antigen presentation mediated by residual leukemic cells not detected by routine screening techniques. It is tempting to speculate that exhaustion of early-differentiated T cells might curtail the long-term reservoir of leukemia-reactive donor T cells, thus facilitating disease relapse. In accordance with this hypothesis, we observed that most tumor-specific T cells retrieved in patients prone to relapse, displayed an exhausted phenotype, whereas the same tumor specificities identified in CR patients were functionally fit and did not express IRs.

The T-cell exhaustion phenotype was challenged in functional assays. In relapsing patients, CD8$^+$ BM-T cells display a reduced IL-2 production and degranulation capacity, while retaining the ability to produce TNF-α and IFN-γ, a functional profile compatible with initial signs of exhaustion[39]. Notably, in vitro activation with high doses of IL-2 reverted the functional defect, as already reported in melanoma patients[26].

A recently published work reported the increased coexpression of inhibitory molecules on circulating T cells recognizing minor histocompatibility antigens after allo-HSCT[40]. Of notice, in our experimental setting, IR$^+$ T cells infiltrating the BM of AML patients at relapse display a skewed TCR repertoire and, after in vitro expansion, in the absence of any procedure aimed at enriching for leukemia-specific T cells, displayed a greater ability to recognize matched leukemic blasts, compared with the IR$^-$ counterparts, indicating that IR expression marks lymphocytes enriched for tumor specificity.

The success of antibodies blocking immune checkpoints in oncology led to their approval for the cure of advanced solid tumors and of relapsed Hodgkin lymphoma as the first hematological indication[41]. Our study provides a rationale for the treatment and prevention of AML post-transplant relapses with multiple immune checkpoints-blocking agents to restore the function of leukemia-reacting T cells. Due to the broad expression of multiple IRs on T cells after transplantation, and the risk of exacerbating GvHD by blocking all relevant immunomodulatory axes in the entire T-lymphocytes population, innovative cell manipulation procedures should be explored to design specific leukemia-reactive cellular products, able to counteract the immunosuppressive leukemic microenvironment.

## Methods

**Study design and human biological samples.** In this retrospective study, we considered 32 leukemic patients who underwent allogeneic hematopoietic stem cell transplantation. Inclusion criteria were a diagnosis of acute myeloid leukemia or myelodysplastic syndrome, a relapse-free survival of at least 4 months after HSCT, absence of moderate-to-severe active GvHD, CMV infections, or other complications at the time of sampling. Relapsing patients fulfilling these criteria were initially identified. Since those patients received HSCT under a variety of treatment schedule[42–44], we selected control patients showing similar clinical characteristic who maintained a status of long-term complete remission (follow-up > 3 years) to avoid late events of disease recurrence and misinterpretation of the results. Samples from relapsing patients were collected at relapse. Patients who maintained CR were evaluated at matched time points. Bone marrow samples from 18 healthy donors were analyzed in parallel as control. For the analyses at early time point, we recruited 57 patients (20 prone to relapse, 37 destined to long-term CR), at a median of 68 (range 28–180) days after HLA-identical HSCT.

PBMC and bone marrow samples were retrieved from the BioBank facility at San Raffaele, at the University Hospital Carl Gustav Carus Dresden, at the Leiden University Medical Centre (LUMC), at the University of Freiburg Medical Center, and at the Hokkaido University Faculty of Medicine where they were collected under written informed consent in agreement with the Declaration of Helsinki. Sample availability is limited and their use is under control and approval from the Ethical Committee of each of the participating institutes.

**Multiparametric flow cytometry.** PBMC and bone marrow samples were thawed and kept overnight at 37 °C at a concentration of $1 \times 10^6$ cells/ml in complete X-vivo: X-vivo (Lonza) supplemented with glutamine 1%, penicillin/streptomycin 1%, human serum 2%, DNase (Roche) 1 IU/ml, and IL-2 20 IU/ml (Novartis). Samples were then stained with mouse fluorochrome-associated monoclonal antibodies specific for human CD3-BV510 (BioLegend, clone OKT3), CD4-PE/Dazzle (BioLegend, clone RPA-T4), CD8-BV785 (BioLegend, clone RPA-T8), CD25-APC/Cy7 (BioLegend, clone BC96), CD45RA-AF700 (BioLegend, clone HI100), CD45RA-PerCP/Cy5.5 (BioLegend, clone HI100), CD62L-AF700 (BioLegend, clone DREG-

**Table 2 Patients' characteristics at early time point follow-up**

| Number of patients, n | | Long-term complete remission (CR) | Relapse (REL) |
|---|---|---|---|
| | | 37 | 20 |
| Diagnosis, n (%) | | | |
| AML | | 33 (89%) | 18 (90%) |
| MDS | | 4 (11%) | 2 (10%) |
| Donor type, n (%) | | | |
| HLA-matched sibling | | 29 (78%) | 19 (95%) |
| HLA-matched MUD (9–10/10) | | 8 (22%) | 1 (5%) |
| CMV serostatus donor/recipient, n (%) | | | |
| N.A. | | 3 (8%) | 1 (5%) |
| pos/pos | | 30 (81%) | 18 (90%) |
| pos/neg | | 2 (5%) | 1 (5%) |
| neg/pos | | 1 (3%) | 0 (0%) |
| neg/neg | | 1 (3%) | 0 (0%) |
| Disease status at transplant, n (%) | | | |
| N.A. | | 3 (8%) | 1 (5%) |
| Complete remission | | 29 (78%) | 15 (75%) |
| Presence of disease | | 5 (14%) | 4 (20%) |
| Conditioning regimen, n (%) | | | |
| N.A. | | 3 (8%) | 1 (5%) |
| Reduced-intensity | | 2 (5%) | 6 (30%) |
| Myeloablative, treosulfan-based | | 8 (22%) | 3 (15%) |
| Myeloablative, other | | 24 (65%) | 10 (50%) |
| In vivo T-cell depletion, n (%) | | | |
| N.A. | | 3 (8%) | 3 (15%) |
| None | | 15 (41%) | 3 (15%) |
| ATG | | 3 (8%) | 2 (10%) |
| PT-Cy | | 14 (38%) | 12 (60%) |
| ATG/PT-Cy | | 0 (0%) | 0 (0%) |
| Other | | 2 (5%) | 0 (0%) |
| GvHD prophylaxis, n (%) | | | |
| N.A. | | 1 (3%) | 1 (5%) |
| CSA-based | | 16 (43%) | 8 (40%) |
| Sirolimus-based | | 10 (27%) | 6 (30%) |
| Other | | 10 (27%) | 5 (25%) |
| Acute GvHD incidence, n (%) | | | |
| N.A. | | 4 (11%) | 1 (5%) |
| Grade 0-I | | 30 (81%) | 18 (90%) |
| Grade II-IV | | 3 (8%) | 1 (5%) |
| Chronic GvHD incidence, n (%) | | | |
| None | | 34 (92%) | 19 (95%) |
| Mild | | 2 (5%) | 1 (5%) |
| Moderate | | 1 (3%) | 0 (0%) |
| Severe | | 0 (0%) | 0 (0%) |
| Under treatment for GvHD at the time of sampling, n (%) | | | |
| None | | 34 (92%) | 16 (80%) |
| Yes | 0–1 mg/kg steroid | 3 (8%) | 4 (20%) |
| Yes, other | | 0 (0%) | 0 (0%) |
| Time point sampling (days after HSCT), mean ± SD (range) | | 71 ± 35 (28–180) | 66 ± 39 (28−174) |

*AML* acute myeloid leukemia, *MDS* myelodysplasia, *ATG* anti-thymocyte antibodies, *PT-Cy* post-transplant cyclophosphamide, *GvHD* graft-versus-host disease, *HSCT* haematopoietic stem cell transplant, *CMV* cytomegalovirus

(BioLegend, clone WD1928), 41BB-AF700 (BioLegend, clone 4B4-1), CD27-FITC (BD Pharmingen, clone M-T271), CD28-PE (BD Pharmingen, clone L293), and CTLA-4-PE (BD Pharmingen, clone BNI-3) and for rat fluorochrome-associated monoclonal antibodies specific for TIM-3-AF488 (R&D, clone 295D) and goat fluorochrome-associated monoclonal antibodies specific for LAG-3-PE (R&D, clone Leu23-Leu450). In addition, for the study of leukemic blasts, samples at relapse were stained with the following fluorochrome-associated monoclonal antibodies: CD11b-BV785 (BioLegend, clone ICRF44), CD34-PB (BioLegend, clone 581), CD45-PE/Dazzle (BioLegend, clone HI30), CD48-FITC (BioLegend, clone BJ40), CD70-PerCp/Cy5.5 (BioLegend, clone 113-16), CD80-Pe/Cy7 (Bio-Legend, clone 2D10), CD117-BV510 (BioLegend, clone 104D2), Galectin-9-APC (BioLegend, clone 9M1-3), OX40L-Biotin (BioLegend, clone ACT35), PD-L1-PE/Cy7 (BioLegend, clone 29E.2A3), PD-L2-PE (BioLegend, clone 24 F.10C12), 41BBL-PE (BioLegend, clone 5F4), CD86-FITC (BD Pharmingen, clone FUN-1), streptavidin-APC/Cy7 (BD Pharmingen), ICOSL-APC (R&D, clone 136726), and CD33-AF700 (eBiosciences, clone WM-53). Samples were incubated with antibodies for 15 min at 4 °C and washed with phosphate-buffered saline (PBS) containing 1% FBS. Fixation and permeabilization were required before CTLA-4[45], FoxP3, T-bet, and Eomes staining and was performed using FoxP3 Fix/Perm buffer, FoxP3 Perm buffer, and Perm Wash Buffer (Biolegend) according to the manufacturer instructions. Flow-cytometry data were acquired using BD LSRFortessa cell analyzer and visualized with FlowJo software (TreeStar).

For the analysis of the antigen-specific CD8[+] T cells, staining with dextramers was preceded by 15 min staining with a Zombie Aqua Fixable Viability kit (Biolegend) and a round of washing (5 mins 1,500 rpm). The cell pellet was then resuspended in 50 μL of PBS containing 10% FBS and the specific dextramer added, namely dextramer Immudex anti-EZH2 HLA*0201 (WB3257-PE), anti-WT1 HLA*0201 (WB3787-PE), anti-WT1 HLA*0201 (WB3469-APC), and anti-PRAME HLA*0201 (WB3312-APC). After 30 min at 4 °C, the samples were washed and then stained 20 min at 4 °C with CD3-V500 (BioLegend, clone SK7), CD4-BUV395 (BioLegend, clone SK3), CD8-BUV737 (BD Biosciences, clone SK1), PD-1-BV650 (BioLegend, clone EH12.2H7), TIM3-AF488, LAG3-BV605 (BioLegend, clone 11C3C65), KLRG1-BV785 (BioLegend, clone 2F1-KLRG1), 2B4-PerCP-Cy5.5 (BioLegend, clone C1.7), Zombie Violet Viability Dye (Biolegend), CD62L-A700, CD45RA-PeCy7 (BioLegend, clone HI100), CD95-PB. Catalogue numbers and dilutions of each fluorochrome-conjugated monoclonal antibody are reported in Supplementary Table 1.

**CD107a degranulation and cytokines production assays**. After thawing and overnight incubation at 37 °C in complete X-vivo, cells were plated at the concentration of $1 \times 10^6$ cells/ml in RPMI 1640 (Lonza) supplemented with glutamine 1%, penicillin/streptomycin 1%, fetal bovine serum 10%, IL-2 20 IU/ml, and in the presence of an inhibitor of metalloprotease activity (TAPI-2 200 microM[46]). After 1-h incubation at 37 °C, GolgiStop (0.66 microl/ml, BD biosciences) and the fluorochrome-conjugated antibody CD107a-FITC (BD Biosciences) were added; in the stimulated wells only, PMA (50 ng/ml; Sigma) and Ionomycin (1 μg/ml; Sigma) were added. After 6 h at 37 °C, cells were first stained with fluorochrome-associated monoclonal antibodies specific for surface molecules, CD3-BV510, CD4-PE/Dazzle, CD8-BV785, CD45RA-PerCP/Cy5.5, CD62L-AF700 (Biolegend); next, cells underwent fixation and permeabilization for intracellular staining with monoclonal antibodies specific for the following cytokines: TFN-α-PE/Cy7, IFN-γ-APC/Cy7, and IL-2-PB (Biolegend). Catalogue numbers and dilutions of each fluorochrome-conjugated monoclonal antibody are reported in Supplementary Table 1. Flow-cytometry data were acquired using BD LSRFortessa cell analyzer and visualized with FlowJo software (TreeStar). Statistical analyses were performed with Prism5 software or later versions (GraphPad). Pie charts with arcs were created using SPICE software[47].

**BH-SNE analysis**. BH-SNE is a dimensionality reduction algorithm that allows low-dimensional embedding of experimental data where each high-dimensional object (i.e., an event associated with many variables, such as a cell stained with many different fluorochromes) is collapsed in 2D space as a low-dimensional point associated to only two variables, BH-SNE1 and BH-SNE2. Such low-dimensional points are positioned in maps in a way that nearby points share similar high-dimensional profiles whereas distant points are dissimilar. BH-SNE runs on a Matlab plug-in called "_Cyt", a visualization platform developed by Dana Pe'er's lab of Computational Systems Biology[36]. BH-SNE algorithm analysis settings were perplexity = 30.00 and theta = 0.5, as suggested by the developers. For the analysis in Fig. 2, the algorithm was applied to downscaled (7500 events/sample) bone marrow CD3[+] events for both panels designed for flow-cytometry analysis, one including PD-1, LAG-3, KLRG1, and 2B4 (panel 1) and the other PD-1, TIM-3, and CTLA-4 (panel 2); for the analysis reported in Fig. 6, the algorithm was applied to downscaled (10,000 events/sample) bone marrow CD8[+] CD3[+] events from a single panel containing fluorophores specific for GITR, HLA-DR, TIM3, LAG3, 2B4, KLRG1, and PD-1. K-means algorithm was directly applied on BH-SNE1 and BH-SNE2 variables and it converged in 140–190 interactions, dividing the biaxial plot into 200 discrete areas (meta-clusters) for each BH-SNE analysis. Meta-clusters were studied by means of percentage, and then RFI was calculated and compared with the threshold for positivity defined from FlowJo analysis.

56), CD62L-APC/Cy7 (BioLegend, clone DREG-56), CD95-PB (BioLegend, clone DX2), CD95-PE/Cy7 (BioLegend, clone DX2), FOXP-3-AF647 (BioLegend, clone 259D), ICOS-PB (BioLegend, clone C398.4A), KLRG1-FITC (BioLegend, clone 2Fi/KLRG1), OX40-APC (BioLegend, clone ACT35), PD-1-PE/Cy7 (BioLegend, clone EH12.2H7), 2B4-APC (BioLegend, clone CI.7), GITR⁻BV711 (BioLegend, clone 108–17), T-bet-BV605 (BioLegend, clone 4B10), Eomes-APC/Cy7

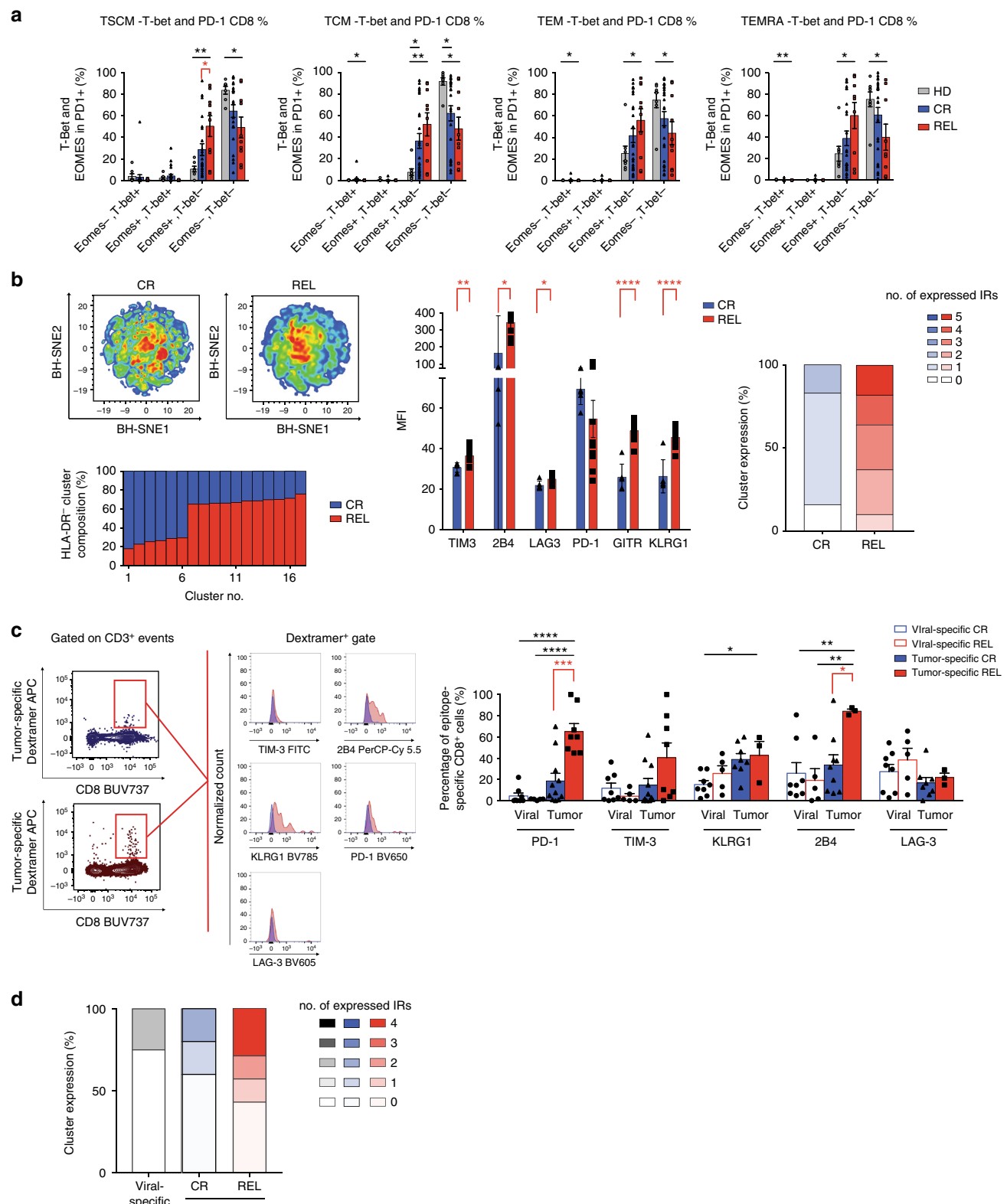

**Isolation and expansion of IR-expressing T cells**. For the isolation IR+ and IR− cells and blasts, thawed bone marrow-derived mononuclear cells were stained with fluorochrome-conjugated antibodies, CD3-BV510, CD34-PB or CD33-AF700, PD-1-PE/Cy7, 2B4-APC, TIM-3-AF488. IR+ cells were defined as CD3+ cells expressing one or more IR according to the gating strategy shown in Fig. 5a. IR+ and IR− (CD3+PD-1−2B4−TIM-3− cells) were sorted accordingly on a Moflo XDP cell sorter (Beckman Coulter). From the same samples, leukemic blasts, defined as

CD34+CD3− or CD33+CD3−, according to the leukemia phenotype, were also retrieved. For all subsets, average purity was greater than 95%. Immediately after sorting, blasts were frozen whereas IR+ and IR− lymphocytes underwent a round of in vitro activation and expansion, as previously described[25]. Briefly, 0.1–0.5 × 10^6 IR+ and IR− T cells were activated with anti-CD3 OKT3 antibody (30 ng/ml, Miltenyi) in RPMI (Lonza) with FBS 10%, glutamine 1%, penicillin/streptomycin 1%, supplemented with recombinant human IL-2 (600 UI/ml, Novartis), and in the

**Fig. 6** A phenotype of exhaustion characterizes tumor-specific T cells early after transplant. **a** Proportion of PD-1[+] CD8[+] BM-T cells expressing T-bet and EOMES transcription factors in different T-cell subsets of healthy donors (HD, $N = 8$), patients who will maintain long-term complete remission (CR, $N = 37$) or will experience disease relapse (REL, $N = 20$). **b** BH-SNE analysis of CR and REL CD8[+] BM-T cells at this early time point. The BH-SNE1/BH-SNE2 biaxial plot for CR and REL and the composition of HLA-DR[−] clusters are reported (left), together with the MFIs of inhibitory receptors (center) and the percentages (right) of CD8[+] BM-T cells expressing inhibitory receptors in the CR- and REL-specific HLA-DR[−] clusters. **c** Representative plot of inhibitory receptors expression by CR or REL tumor-specific CD8[+] T cells and their quantification by histograms, compared with viral-specific CD8[+] T cells. **d** Percentages of expression of multiple inhibitory receptors by viral and tumor-specific T cells. Individual patient data points, means, and SEM are shown. Statistically significant differences between CR and REL groups are highlighted in red and the differences between patients' groups and HD in black. *$p < 0.05$, **$p < 0.01$, ***$p < 0.001$, ****$p < 0.0001$, nonparametric unpaired two-sided $T$ test

presence of irradiated (30 Gy) allogeneic PBMCs ($1 \times 10^6$ cell/ml) obtained from a pool of three different healthy donors, and irradiated (100 Gy) lymphoblastoid cell lines ($0.2 \times 10^6$ cell/ml). The medium and IL-2 were changed every 3–4 days, cells counted by Trypan Blue exclusion and resuspended at a concentration of $0.5–1 \times 10^6$ cells/ml. When expansion reached plateau, IR[+] and IR[−] cells were functionally analyzed in a CD107a degranulation assay, in a cytokine release assay and a co-culture assay. Catalogue numbers and dilutions of each fluorochrome-conjugated monoclonal antibody are reported in Supplementary Table 1.

**Co-culture assay**. IR[+] and IR[−] cells were separately cultured for 3 days in U-bottom 96 wells in complete X-Vivo plus IL-2 60 IU/ml, IL-3 20 ng/ml, and G-CSF 20 ng/ml, in the presence of matched leukemic blasts, at increasing effector-to-target (E:T) ratios (1:10, 1:1, 10:1, 50:1, 100:1). As an internal control, leukemic blasts were cultured in the presence of unrelated PBMCs. After 24 h, supernatants were collected and, after a cycle of freezing and thawing, processed using LEGENDplex Human CD8/NK panel (Biolegend) and read at BD FACSCanto II HTS; this panel allows to measure molecules released upon activation, including Granzyme A and Granzyme B, in the co-culture supernatant. After 3 days of coculture, the total number of target cells was quantified upon staining with Zombie Aqua Viability kit (Biolegend), with fluorochrome-conjugated antibodies specific for CD3, CD4, CD8, CD33, or CD34 (according to the leukemia phenotype), CD45, CD117, HLA-DR-FITC (Biolegend). Data were acquired at BD LSRFortessa cell analyzer. Viable target cells were counted by using counting beads (Flow-count fluorospheres, Beckman Coulter), according to the manufacturer instructions. Elimination index was calculated according to the formula: [1− (total number of residual target cells cultured in presence of IR[+] or IR[−] T cells)/(total number of residual target cells cultured in presence of unrelated PBMCs)] × 100. Catalogue numbers and dilutions of each fluorochrome-conjugated monoclonal antibody are reported in Supplementary Table 1.

**Analysis of TCR-α and TCR-β chains repertoire**. Sequencing of the TCR-α and -β chains was performed on RNA by using a modified RACE PCR protocol, independently of multiplex PCRs[48,49]. Samples were sequenced using the MiSeq platform (Illumina) and raw reads were sorted according to the individual barcode combination used for each specimen. Analysis of the TCR clonotypes was carried out using the MiXCR software[50].

**Statistical analysis**. Nonparametric unpaired two-sided $T$ tests were performed for the analysis of the set of data throughout the study, with the exception of Figs 3b, 5d, where the comparison of the variables from three different experimental groups was carried out by two-way ANOVA with no matching coupled with Sidak multiple correction test, and Fig. 5g, where the comparison between the two experimental T-cell subsets IR[+] and IR[−] isolated from the same patients at the same time point was compared by means of paired two-tail T test.

Statistical analyses were performed with Prism5 software or later versions (GraphPad). The heatmap of Fig. 2 was created by using Rstudio software, packages "pheatmap", "ggplot2", and "RColorBrewer".

## Data availability
The TCR sequences data that support the findings of this study are available in SRA database with the identifier PRJNA510967. Further data are available from the corresponding author on reasonable request.

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

## Acknowledgements

This work was supported by EU-FP7 (SUPERSIST) to C.B., by the Italian Association for Cancer Research (AIRC-IG-18458 to C.B., Start-Up Grant #14162 to L.V.), by the Italian Ministry of University and Research (MIUR-2015NZWSEC_001) to C.B., the Italian Ministry of Health (RF-FSR-2008-1202648 to Fabio Ciceri, RF-2011-02351998 to Fabio Ciceri and L.V., RF-2011-02348034 to L.V. and TRANSCAN HLALOSS to L.V.), by the ASCO Conquer Cancer Foundation (2014 Young Investigator Award to L.V.), and by the DKMS Mechtild Harf Foundation (DKMS Mechtild Harf Research Grant 2015 to L.V.). E.R. was supported by a fellowship from the European Union's Horizon 2020 research and innovation program under the Marie Skłodowska-Curie grant agreement THAT IS HUNT 752717. This work was partially supported by the Italian Ministry of Health (GR-2016-02364847) to E.R.

## Author contributions

M.N. designed the study, conducted laboratory experiments, analyzed and interpreted data, and wrote the paper; F.M. conducted laboratory experiments and BH-SNE analysis, analyzed and interpreted data, and wrote the paper; E.R. performed and analyzed experiments of TCR sequencing; T.P., R.G., and J.P. provided clinical data and participated in the data interpretation; G.O. and N.C. participated to the design of the study and data discussion; Filippo Cortesi participated to the BH-SNE analysis; P.D.S. participated to laboratory experiments; C.T. and V.G. participated in the data interpretation; M.C. participated to co-culture experiments and data discussion; G.C. and P.D.B. participated to the interpretation of results and reviewed the paper; M.O., T.T., G.M., H.C.J.M., F.J.H.F., S.F., H.H., B.M., M.W., R.Z., and J.F. provided clinical data and bone marrow samples, and participated in data interpretation; A.B. participated to data discussion and interpretation of the results; L.V., Fabio Ciceri, and C.B. designed and supervised the study and wrote the paper.
