## [Peer Review File · Nature Communications]

REVIEWERS' COMMENTS:

Reviewer #2 (Remarks to the Author):

I would like to thank the authors for responding thoroughly to the reviewers' queries

Reviewer #3 (Remarks to the Author):

1) Please clarify in the text whether the PD-1+ Eomes+ exhausted TSEM cells detected are in the HLA DR+ set, as inferred from "activated immunological setting", or in the HLA DR- set.

2) The data in Figure 6C are of high interest. Are the PD-1+ . Tim 3+ and 2B4+ tumor specific T-cells isolated by dextramers HLA DR+ or DR-, or was this defined.

3) In the discussion, the authors might briefly address whether they consider the oligoclonal exhausted TSCM deleted in the bone marrow early after transplant that are leukemia-specific reflect represent populations of T-cells "spent" on terminally exhausted as a result of responses against residual leukemia cells not detected by current techniques that mark the relapse -prone individual or rather, reflect a very limited repertoire of potentially leukemic-reactive T-cells available in transplants from specific donors.

Point-by-point reply to Reviewers' comments:

Reviewer #2 (Remarks to the Author):

I would like to thank the authors for responding thoroughly to the reviewers' queries

Reviewer #3 (Remarks to the Author):

1) Please clarify in the text whether the PD-1+ Eomes+ exhausted TSCM cells detected are in the HLA DR+ set, as inferred from "activated immunological setting", or in the HLA DR- set.

We clarified in the text that the visualization of PD-1+ Eomes+ TSCM have been performed from the total CD8+ T cell repertoire resident in the Bone Marrow.

2) The data in Figure 6C are of high interest. Are the PD-1+, Tim 3+ and 2B4+ tumor specific T-cells isolated by dextramers HLA DR+ or DR-, or was this defined.

Results of Figure 6C show Dextramer+ T cells visualized on bulk CD8+ T cells, without previous gating on HLA-DR+ or HLA-DR- subpopulations. However, in a different set of experiments, we observed that the expression of HLA-DR from tumor-specific T cells harvested from either CR and REL patients is comparable.

3) In the discussion, the authors might briefly address whether they consider the oligoclonal exhausted TSCM deleted in the bone marrow early after transplant that are leukemia-specific reflect represent populations of T-cells "spent" on terminally exhausted as a result of responses against residual leukemia cells not detected by current techniques that mark the relapse-prone individual or rather, reflect a very limited repertoire of potentially leukemic-reactive T-cells available in transplants from specific donors.

It's plausible to assume that the exhausted phenotype that we described is the result of an immune pressure exerted by the tumor microenvironment and by the leukemia itself (namely the Minimal Residual Disease), redirecting (or freezing) the T cell repertoire into a phenotype, poorly adapted to cytotoxic functions And characterized by the expression of the variety of inhibitory receptors.

Our study cannot rule out that the phenotype we described is the result of donor-derived T cell clones already present in the graft with an exhaustion phenotype, later expanding in the host because of antigen encounter. However, the requirement of tonic antigen stimulation with limited costimulation for exhaustion, associated to the low proliferation capacity of exhausted T cells, suggest that this phenotype is actually acquired by T cells upon transplant.

The sentence "The generation of such exhausted T cells at an early time point might be the result of a continuous aberrant antigen presentation mediated by residual leukemic cells not detected by routine screening techniques." has been added to the manuscript in the Discussion session to clarify this point.